# An Account of Acute Myeloid Leukemia Complicating Pregnancy and Literature Review

**DOI:** 10.3390/diagnostics15192540

**Published:** 2025-10-09

**Authors:** Georgiana Nemeti, Laura Jimbu, Oana Mesaros, Iulian Gabriel Goidescu, Cezara Moisa, Mihai Surcel, Cerasela Mihaela Goidescu, Dan Boitor-Borza, Gheorghe Cruciat, Ioana Cristina Rotar, Daniel Muresan

**Affiliations:** 1Obstetrics and Gynaecology I, Mother and Child Department, University of Medicine and Pharmacy “Iuliu Hațieganu”, 400006 Cluj-Napoca, Romania; georgiana.nemeti@elearn.umfcluj.ro (G.N.); mihai.surcel@elearn.umfcluj.ro (M.S.); dan.boitor@elearn.umfcluj.ro (D.B.-B.); gheorghe.cruciat@elearn.umfcluj.ro (G.C.); cristina.rotar@umfcluj.ro (I.C.R.); muresandaniel01@elearn.umfcluj.ro (D.M.); 2Department of Haematology, Ion Chiricuta Oncology Institute, Iuliu Hatieganu University of Medicine and Pharmacy, 400006 Cluj-Napoca, Romania; ioana.jimbu@umfcluj.ro (L.J.); mesaros.oana@umfcluj.ro (O.M.); 3University of Medicine and Pharmacy “Iuliu Hațieganu”, 400006 Cluj-Napoca, Romania; moisa.cezara.andrea@elearn.umfcluj.ro; 4Department of Internal Medicine, Medical Clinic I—Internal Medicine, Cardiology and Gastroenterology, University of Medicine and Pharmacy “Iuliu Hațieganu”, 400006 Cluj-Napoca, Romania; sava.cerasela@elearn.umfcluj.ro

**Keywords:** acute myeloid leukemia, pregnancy, azacitidine, delivery

## Abstract

**Background and Clinical Significance:** The occurrence of acute myeloid leukemia (AML) in pregnancy represents a diagnostic and management challenge in the attempt to balance and achieve both maternal and fetal wellbeing. Pregnancy-specific manifestations mimic the initial symptoms of leukemia and may lead to a delay in diagnosis, especially during the first trimester of pregnancy. Decision-making strategies involve the patient and couples counseling with a multidisciplinary team of hematologists, obstetricians, neonatologists and psychologists. Maternal outcome depends on the disease subtype, progression and response to medication. Fetal outcome depends on other potential pregnancy complications, possible teratogenicity, gestational age at delivery and sometimes iatrogenic prematurity. **Case Presentation:** We present the case of a 38-year-old multiparous patient with a late first trimester, with an AML diagnosis presenting with hyperemesis gravidarum-like symptoms. Genetic testing revealed the presence of an Fms-like tyrosine kinase 3-internal tandem duplication mutation (FLT3-ITD). Following that, a repeatedly refused termination of pregnancy and rapid disease progression with azacitidine therapy was initiated. Elective cesarean delivery was performed at 34 weeks of gestation due to progressive leukocytosis, which persisted postpartum, requiring the use of first-, second-, and eventually third-line chemotherapy. Fetal outcome was favorable at 3 months postpartum. **Conclusions:** Cases of AML in pregnancy require a tailored approach according to guidelines, but also patient/couple preferences, while the choice of chemotherapy is limited considering its potential teratogenic effects. This is a case with a misleading first presentation and a challenging therapeutic choice due to its genetic subtype and maternal treatment postponement.

## 1. Introduction

Even though the incidence of cancers diagnosed during pregnancy is reportedly 0.07% to 0.1%, they represent the second most frequent cause of maternal death following obstetric haemorrhagic complications [1,2]. When compared to same-age non-pregnant females, pregnant patients were reportedly diagnosed with the same cancer sites and similar frequencies [3,4,5]. Among hematologic cancers complicating pregnancy, leukemia was diagnosed in 1 in 75,000 to 100,000 pregnancies, with acute myeloid leukemia (AML) accounting for approximately 60% of cases [1,3,4,5,6]. AML is characterized by the clonal proliferation of ≥20% poorly differentiated white blood cells, called blasts [7]. Based on a better understanding of the genetics of AML, currently, in the presence of certain abnormalities, the standard cut-off of 20% blasts has been lowered to 10% to achieve diagnosis [8]. Most cases of AML are diagnosed during the second and third trimesters of pregnancy, with roughly 23% picked up during the first trimester [3,6].

Diagnosing such a rare pathology is tricky because physiological changes in pregnancy often mimic some of the symptoms related to the clinical presentation of leukemia, such as fatigue, anemia, thrombocytopenia, nausea and weakness [8]. The first clues towards diagnosis are provided by changes in the mandatory blood tests performed during pregnancy—such as severe neutropenia, anemia, thrombocytopenia, leucocytosis or leukopenia—and the presence of more than 20% blasts in the bone marrow or peripheral blood. According to the International Consensus Classification of Myeloid Neoplasms and Acute Leukemias, AML can also be defined as ≥10% blasts in the presence of certain genetic abnormalities, which can be found at https://ashpublications.org/blood/article/140/11/1200/485730/International-Consensus-Classification-of-Myeloid (accessed on 20 September 2025) [6]. The diagnosis is often confirmed by bone marrow aspiration, which reveals an elevated number of blast cells by flow cytometry, confirming the myeloid lineage.

The coexistence of AML and pregnancy presents a challenging context starting from the initial presentation, and diagnosis requires a multidisciplinary team (hematologist, obstetrician and neonatologist) to establish the management protocol and to counsel patients regarding both maternal and offspring outcome. The presence of the fetus precludes the use of chemotherapy during organogenesis—the first trimester—and limits the choice of drugs thereafter, considering the risks of growth restriction, stillbirth or neonatal death, prematurity and myelosuppression.

The management of AML during gestation is a challenging process due to the limited use of chemotherapy, but also because maternal complications—both disease-specific or pregnancy-specific—are more difficult to manage. At the same time, since cases are not that frequent and there is great variation among them regarding maternal biologic status, gestational age at diagnosis, clinical and biological disease characteristics, maternal/fetal drug adverse effects, maternal/couple understanding and treatment options, it is difficult to obtain a large series of cases and conduct standardized guidance [9,10].

We present the case of an AML patient diagnosed during the first trimester of pregnancy, who was repeatedly counseled to terminate pregnancy but refused and was then managed multidisciplinarily. On this basis, we conducted a review of the existing literature regarding the diagnostic and management possibilities for such cases.

## 2. Case Report

This is the case of a 38-year-old G IV P IV patient with a confirmed pregnancy admitted at 11 weeks of gestation (WGs) for complaints of extreme tiredness, nausea and vomiting. The clinical examination was unremarkable except for the identification of gingival hyperplasia. In the context of pregnancy, the case was interpreted as hyperemesis gravidarum, and supportive management, antiemetics, fluids and electrolytes were started. The complete full blood count revealed hyperleukocytosis (190.79 × 10^9^/L) with monocytosis (95.86 × 10^9^/L), neutrophilia (58.17 × 10^9^/L) and lymphocytosis (36.60 × 10^9^/L), severe anemia (hemoglobin = 6.8 g/dL) and thrombocytopenia (38 × 10^9^/L). The peripheral blood smear revealed 77% blasts with Auer rods. Immunophenotyping showed myeloblasts positive for CD33, CD34 partially, CD117, HLADR, CD13, CD7, CD38 and negative for CD11b, CD14, CD56, CD19, CD22 and nuclear terminal deoxynucleotidyl transferase (nuTdt). An acute myeloid leukemia diagnosis was established. Upon genetic testing, the patient was positive for the mutation Fms-like tyrosine kinase 3-internal tandem duplication (FLT3-ITD), negative for Nucleophosmin 1(NPM1) and BCR-ABL (a balanced reciprocal translocation between the long arms of chromosomes 9 and 22). The patient’s karyotype was normal, at 46XX (Figure 1).

Considering that the gestational age at diagnosis was during the first trimester, the need to employ chemotherapy for proper disease control, in adjunction with the high risk of feto–maternal complications, the patient and her spouse were counseled to terminate pregnancy, but they refused on several instances.

The 3+7 regimen (cytarabine: 200 mg/m^2^ on days 1–7, continuous infusion + idarubicin: 12 mg/m^2^ days 1–3) is the standard treatment for treating young, fit AML patients, and it has been used as such since 1973. A FLT3 inhibitor is associated with the standard regimen when a mutation in this gene is present. Less often, FLAG-Ida is used in the first line [11].

Intensive chemotherapy (3+7 regimen) was taken into consideration, but when the possible side effects were presented to the patient, she deferred aggressive chemotherapy, especially due to her fear of fetal death. There is no contraindication to treating AML patients with intensive chemotherapy during pregnancy, especially in the second and third trimesters. Several studies have presented that 3+7 is safe to be administered during pregnancy; however, it is associated with a risk of different complications, ranging from skeletal abnormalities to cardiomyopathy or stillbirth [12,13].

Intensive chemotherapy is also associated with several maternal toxicities such as infection, bleeding or even death. In our department, historically, early death (death in the first month from diagnosis) was about 17% [14].

On the other hand, azacitidine is less used as a single agent in the era of venetoclax for AML patients. However, there are several case reports that presented the safe and efficacious use of azacitidine in pregnant patients with AML. Thus, this drug can be considered as a bridge to a more aggressive treatment or to a transplant after delivery [13,15]. FLT3 inhibitors are not considered to be safe to be administered during pregnancy.

Knowing all the possible side effects, risks and benefits, the patient chose a low-intensity treatment approach. Azacitidine single-agent therapy was started on the premise that the benefits outweighed the risks. Venetoclax, a B-cell lymphoma 2 (BCL2) inhibitor, was omitted due to the high risk of teratogenicity. Azacitidine belongs to the class of hypomethylating agents and is typically administered in 4–6 cycles, according to the following regimen: 75 mg/m^2^ for 7 consecutive days, followed by a 21-day break [16]. Our patient was treated with five cycles of azacitidine, starting from 15 WGs, with the last cycle administered at 31 WGs.

The patient was managed by close hematologic monitoring with blood transfusions whenever anemia levels required it for both maternal and fetal benefit.

Repeated cardiologic assessments revealed normal cardiac ultrasound appearance and function.

From the obstetrical point of view, fetal surveillance was normal at all prenatal check-ups, including biometry measurements, morphology scanning, Doppler studies and biophysical profile score. At 28 weeks of gestation, gestational diabetes mellitus was diagnosed following OGTT (oral glucose tolerance test) screening and insulin treatment, and both long- and short-acting products were required for glycemic control.

At 34 weeks of gestation, given the progressive leukocytosis (46.14 × 10^9^/L) developing under treatment and the normally developed fetus, the multidisciplinary team of hematologists, obstetricians and neonatologists agreed on the indication of delivery, even in the context of late prematurity. Since blood products could only be ensured for elective birth a scheduled cesarean section was carried out with the delivery of a 2200 g newborn with Apgar scores of 10 at 1, 5 and 10 min.

The postoperative course was uneventful, and the patient was discharged on day 4 postpartum with low-molecular-weight heparin prophylaxis during the entire postpartum period. At 2 weeks postpartum, the patient was started on azacitidine + venetoclax, but without achieving complete remission (CR), and with progressive leukocytosis. After a shared decision with the patient, it was agreed to start second-line treatment with intensive chemotherapy—the “3+7” regimen, unfortunately, was carried out without an FLT-inhibitor due to national insurance regulations. At follow-up, the patient had persistent blasts in the peripheral blood smear, and she was therefore started on the third-line treatment with an FLAG-Ida (fludarabine, cytarabine, idarubicin, granulocyte colony-stimulating factor) regimen, and an allogeneic stem cell transplant was planned if CR would be achieved. At day 15 of treatment, 30% blasts were detected in the peripheral blood smear (Figure 2). At day 30, a bone marrow aspiration was performed, showing 60–65% blasts. Currently, the patient is receiving giltertinib (120 mg), an FLT3 inhibitor (120 mg). Unfortunately, after one month of post-treatment initiation, the patient still had not achieved CR, with persistent blasts in the bone marrow. Table 1 presents the patient laboratory workup at different timepoints during her surveillance throughout pregnancy.

The fetal postpartum outcome was favorable, with only mild, transient neonatal respiratory distress syndrome and adequate weight gain. Blood tests did not reveal any hematologic abnormalities regarding cell count or appearance on the blood smear. Internal organ ultrasound was normal, including echocardiogram. The baby was discharged on the seventh day of life, and reassessed at the 3-month follow-up, when development and neurologic acquisitions were according to expectations, and there were no pathologic findings at check-up analysis and imaging scans.

## 3. Discussion

The presentation of hematologic disorders in pregnancy is rare, and even this fact alone poses the challenge of recognizing the pathology and managing patients. Even though guidance is available, the tailored approach according to disease-specific requirements, patient comorbidities, obstetric context and maternal/couple perspectives makes decision-making challenging and influences fetal and maternal outcomes. Each case is unique in its set-up, and management is based on a multidisciplinary counsel, standardized guidance, same-center experience and previously reported cases in the literature. The caseload of pregnancies complicated by AML during the past 20 years is encompassed in Table 2 since there have been no national or global datasets available.


*AML clinical presentation*


At presentation, AML patients have non-specific symptoms such as asthenia, fatigue, frequent infections in their recent history or unusual bruising or bleeding. The complete blood count usually suggests a bone marrow insufficiency characterized by anemia and thrombocytopenia with either leukopenia or leukocytosis. In care cases, these patients can also present with a tumor mass—myeloid sarcoma—with central nervous system or skin involvement [36].

Anemia-related symptoms can precede patient presentation by months and are represented by fatigue, pallor and weakness. Neutropenia-related infections causing fever is also a common cause of seeking medical care, while bleeding and easy/frequent bruising may be related to thrombocytopenia.

In this context, and while pregnancy itself may present with such nonspecific symptoms in the guise of physiologic changes (fatigue, weakness, pallor, nausea), a delay in AML diagnosis is often recorded and mainly established during the second and third trimesters of pregnancy. Physiological blood parameter changes associated with pregnancy, dilutional anemia of pregnancy, leukocytosis or gestational thrombocytopenia are typically present in AML. Further, recurrent infections and bleeding occurring may be the expression of bone marrow failure [28].


*AML diagnostic considerations*


Establishing the diagnosis of AML requires an integration of clinical, genetic, and pathologic parameters, which allow the proper classification of disease subtype, assessment of risk level and guidance of patient counseling and management. Genetic testing is thus of paramount significance with genes such as NPM1, FLT3 and CCAAT enhancer binding protein alpha (CEBPA), carrying an important impact on patient outcome, especially in patients with a normal karyotype [7].

During pregnancy, the diagnosis of AML is challenging since both clinical and biological parameter alterations may be explained by physiologic maternal adaptive changes. However, the identification of blasts in peripheral blood represents a diagnostic lead, which is then confirmed by bone marrow aspiration and followed by immunophenotyping. Molecular biology studies, karyotyping and, more recently, next-generation sequencing tests are of paramount importance to achieve risk stratification [8].

A peculiar case of AML presentation in advanced pregnancy mimicking appendicitis or abruptio placentae due to severe, continuous abdominal pain was reported [22].

From a genetic point of view, regarding the mutation identified in the patient presented, FLT3/ITD mutation [37] is associated with an intermediate prognosis, based on the ELN (European Leukemia Net) classification [11,38]. However, FLT3-ITD-mutated AML is associated with a [39] more aggressive disease and with a higher risk of relapse. FLT3-ITD is a driver mutation with an impact on DNA methylation, histone modification, and chromatin remodeling, promoting alterations in cellular growth and enhancing disease development [40,41,42]. Malfunctioning epigenetic mechanisms will lead to oncogene activation, tumor suppressor gene inactivation and chromosomal damage, which promote and maintain cancer expansion.

Certain behavioral exposures—smoking, chemicals, prior chemo/radio therapy, exhibiting another myelodysplastic syndrome and inherited conditions such as Fanconi anemia and Diamond Blackfan anemia, and carriers of genetic mutations in the germline, namely runt-related transcription factor 1 (RUNX1), CEBPA and GATA binding protein 2 (GATA2), represent risk factors for AML development [43,44].

A case of genetic pick-up of a maternal abnormality in an otherwise asymptomatic patient at NIPT (non-invasive prenatal testing) was reported, highlighting the importance of such tests, even in maternal interest [45].


*AML management considerations*


AML management is generally guided by the AML subtype as well as corresponding risk assessment, patient age, central nervous system involvement, presence of concurrent infective status, white blood cell counts at treatment initiation, prior chemo/radiotherapy exposure and type/cumulative dose thereof, personal history of myelodysplastic syndrome and personal history of other cancer types. Following patient evaluation, a tailored management comprising chemo/radiotherapy and/or targeted therapies is commenced, with concomitant supportive therapy.

The diagnosis of leukemia during pregnancy, such as any neoplasm, complicates the decision-making process, requires a complex multidisciplinary consultation team including the obstetrician and neonatologist, and limits the therapeutic arsenal to protect the fetus.

The first trimester of pregnancy is special because both the maternal and fetal interface are more vulnerable to outside interference. Looking from the perspective of an AML patient, on the maternal side, the immune system is not yet fully adjusted to the pregnant state and is thus more susceptible to infections, bleeding and pregnancy loss; on the fetal side, the risk of teratogenicity is the highest during this period [8,12,32].

Since the occurrence of AML in pregnancy is exceptional, there are no specific guidelines to establish orderly management, but due to all of the above considerations, pregnancy termination following the first trimester diagnosis of AML is a valid option, which allows for proper maternal hematologic therapy and disease remission [46]. However, medical termination of pregnancy is advocated over chemotherapy, as delaying treatment could often be fatal.

When AML is diagnosed later in pregnancy, during the second or third trimesters, the risks of chemotherapy decrease, even though they are not completely abolished. It may be provided with close surveillance of fetal abnormalities. Chemotherapy is recommended between 13 and 24 WGs, while it is crucial to mention that after 24 WGs, the threat of prematurity outweighs the risks of chemotherapy [12].

As such, Chang et al. in a large series of cases ranging from 1969–2014, report the use of pharmacologic therapy during pregnancy and the rapid initiation for fear of an unfavorable outcome if treatment were delayed [28]. This was especially opted for during the second and third trimesters of pregnancy, when fetal risks were deemed lesser. If AML is confirmed later in the third trimester, after 32 WGs, the patient/couple may be counseled for labor induction and proper hematologic chemotherapeutic management in the postpartum period.

Reports of children born from pregnancies treated for malignancy suggest no consequences, even in their offspring to follow [7,26]. While many infants may be healthy at initial examination, the adverse effects from chemotherapy exposure may not appear until many years after birth, and reports are sparse, with the pool of data needing renewal from more recent analysis. The outcomes of the patient and the fetus must be considered carefully when selecting the chemotherapy regimen used to treat the leukemia. Furthermore, most of the patients received combination chemotherapy, so it is difficult to distinguish which agent may be the cause of certain birth defects or adverse effects in the infant.

From the point of view of the class of chemotherapeutic agents, the preferred therapy during the second and third trimesters is anthracyclines. Doxorubicin is mainly used in pregnancy since idarubicin, as a lipophilic molecule, crosses the placenta. Given the acknowledged risk of cardiotoxicity of anthracyclines, regular fetal cardiac scans are mandated [28]. Cytarabine use is restricted during pregnancy due to the adjunct risk of limb dysplasia, especially during the first trimester, causing pancytopenia, endometrial malfunction, limitation of endometrial development and neonatal death [7]. In a study by Horrowitz et al., complete remission was achieved by anthracycline–cytarabine-based regimens in 91% of patients; however, the maternal overall survival was 30% [26].

Even if in pregnancy the treatment choice and initiation of it are often guided by the gestational age, it is important to consider the specific mutations found in patients. Specifically, in relation to the FLT3-ITD mutation, a conventional 7+3+a FLT3 inhibitor (idarubicin 12 mg/m^2^ on days 1 to 3 and cytarabine 200 mg/m^2^ on days 1 to 7 + midostaurin) is considered the standard treatment [40,45,47,48]. The use of cytotoxic drugs is not strictly contraindicated in the second and third trimester but is associated with multiple risks both for the mother and for the fetus [12]. Thus, the patient did not agree to intensive chemotherapy and wished to pursue a “milder” approach.

Azacitidine, a hypomethylating agent that works by reactivating silenced tumor suppressor genes, is considered a lower toxicity drug as compared to classical chemotherapy agents, offering better outpatient tolerance and compliance [49]. These characteristics are essential for pregnant patients diagnosed with AML, where maternal and fetal stability are critical, and this is why this was an acceptable choice of therapy in our case. Case series were published suggesting that azacitidine could be administered safely in pregnant patients [15,49,50].

Although not commonly used during pregnancy, hydroxyurea should be considered in cases of hyperleukocytosis, as it can control leukemic proliferation until chemotherapy is initiated. Similarly, leukapheresis may serve as a temporary alternative for the same purpose [51].

Acute promyelocytic leukemia (APL) is a rare subtype of AML, characterized by t(15:17) and in contrast with other subtypes of AML, associated with spectacular results. The advent of arsenic trioxide (ATO) combined with all-trans retinoic acid (ATRA) has improved even more the overall survival and CR rate. In some studies, the CR rate of these patients was close to 100% with an event-free survival of 97% [52]. However, real-world data show quite a high rate of early mortality (EM), varying from 20% to 7% [53]. In patients with non-APL AML in EM at our center, those treated with intensive chemotherapy were 17.3%, mainly because of septic complications [14].

APL in pregnancy is rare, with no standardized approach. The use of ATRA and ATO or even idarubicin is associated with teratogenic risks. Table 3 presents data of APL pregnant patients and the outcomes focused, both on the mother and the fetus.


*Pregnancy outcome in AML patients*


The course of AML may be complicated by various factors, primarily hematologic changes that often lead to infections and bleeding due to neutropenia and thrombocytopenia. Organ dysfunction, particularly liver and kidney impairment, is also a complication associated with treatment. Additionally, the worsening of the maternal condition may prompt the decision to escalate to a more aggressive treatment, which cannot be initiated during gestation. The occurrence of such complexities in pregnancies complicated by AML can cloud both fetal and maternal outcomes and may lead to the decision to deliver.

At the same time, delivery might be indicated in the presence of obstetric complications such as intrauterine growth restriction, stillbirth, acute fetal distress, preeclampsia, gestational diabetes and venous thromboembolism [69,70].

In this case, the multidisciplinary team managing the case decided on early delivery at 34 WGs due to the rising trend of leucocytes threatening to turn into a leukemic crisis, which would worsen the maternal state and outcome by making disease control difficult, negatively impact fetal state in utero mandating urgent delivery, while handling a critical maternal status.

The choice of delivery route was elective cesarean section to be able to prepare blood products and ensure a smooth perioperative and perinatal transition for the maternal body. A secondary reason was the lack of nationally available prostaglandin to induce labor at 34 weeks of gestation. The probability of successful induction with oxytocin use would be small at such an early gestational age, even for a higher-rank multiparous patient such as ours.


*Fetal and neonatal outcome in AML patients*


Fetal abnormalities are more likely to occur if chemotherapy is initiated during the first trimester, as the drugs can cross the placental barrier, potentially leading to spontaneous abortion, fetal death or malformations [40,71]. In the second and third trimesters, the primary concerns are preterm delivery and low birth weight. However, long-term follow-up in most studies has shown that the development of newborns born to mothers with AML is typically normal [72].

With regard to the potential vertical transmission of the disease, this situation is extremely rare because the placenta acts as a strong immunologic and physical barrier, but most importantly because AML it is not a genetic disorder, therefore there is no hereditary transmission, unless it is part of a rare genetic syndrome, such as Down syndrome, familial monosomy 7, Fanconi anemia or Diamond-Blackfan anemia [1,8].

Idarubicin has been linked to having fetal cardiotoxic effects, which may result from its biophysical properties facilitating the transplacental passage, high liposolubility and long half-life [45].


*Long-term maternal outcome following AML diagnosis during pregnancy*


The life-long prognosis of a pregnant patient with AML depends primarily on leukemia-specific factors and not the pregnancy itself. They are considered favorable prognosis factors if the patient achieves a CR after therapy and if she has no high-risk cytogenetic features.

Fertility is affected by chemotherapy because it induces apoptosis in primordial follicles, which serve as the reservoir for future fertility. However, the risk of infertility increases with age and dosage, and fertility preservation options counseling should be available for all patients. Younger patients may tolerate larger cumulative drug doses of chemotherapy and still maintain menstruation after a variable period [73,74]. Our patient was 38 years old and delivering her fourth child; therefore, fertility preservation was not included in our counseling strategies.

## 4. Conclusions

This case underscores the challenges inherent in clinical decision-making concerning the optimal timing of therapeutic initiation, the selection of a treatment strategy that ensures both maternal efficacy and fetal safety and the determination of the appropriate timing for delivery—balancing the mother’s progressively deteriorating clinical status with the fetus’s normal intrauterine development relative to gestational age. The optimal approach in this case was to proceed with a premature cesarean delivery, given the high risk of progression to blast crisis, while considering the normal fetal development at 34 WGs.

Unfortunately, the prognosis of this patient is unfavorable, given that she is refractory to two lines of treatment.

## Figures and Tables

**Figure 1 diagnostics-15-02540-f001:**
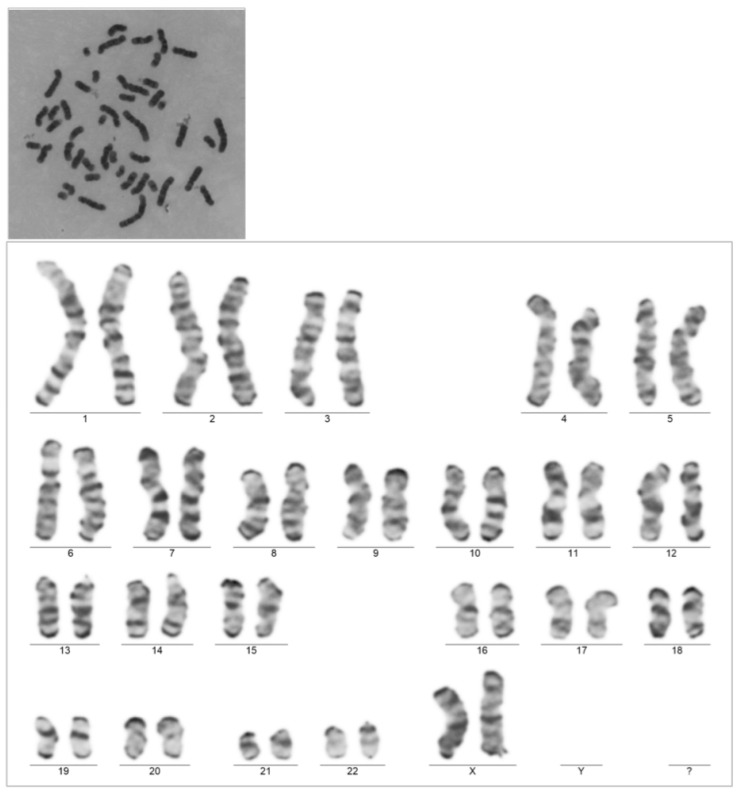
Patient’s karyotype.

**Figure 2 diagnostics-15-02540-f002:**
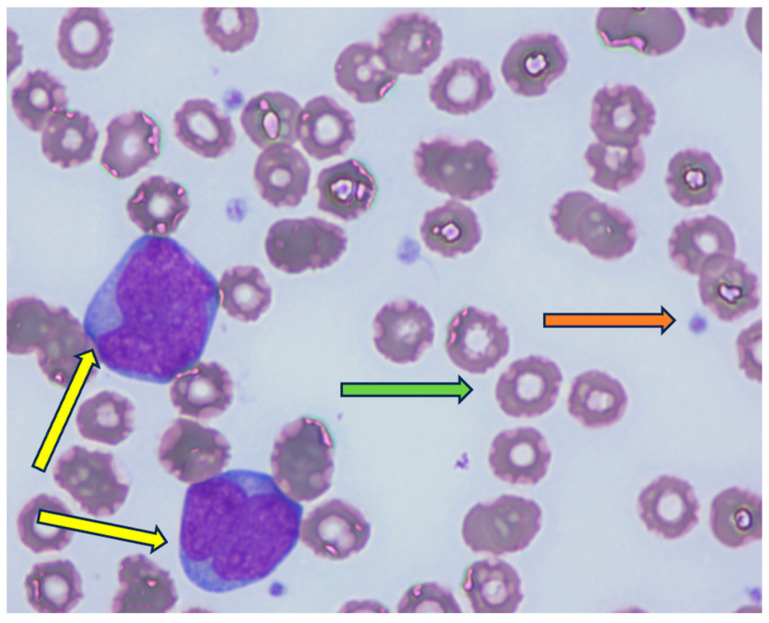
Peripheral blood smear image at day 15 after the FLAG-Ida course (yellow arrow—blasts, green arrow—erythrocyte, orange arrow—platelet).

**Table 1 diagnostics-15-02540-t001:** Parameters from complete full blood count at different timepoints during patient management.

Parameter	Normal Ranges	At Diagnosis (25 October 2024)	Before Starting 3+7 (17 June 2025)	Before Starting FLAG-Ida (15 July 2025)
Leukocytes (×10^3^/μL)	4.49–12.68	190.79	92.61	117.90
Hemoglobin (g/dL)	11.9–14.6	6.8	7.2	9.9
Platelets (×10^3^/μL)	150–450	38	240	175
Peripheral Blasts (%)	0	77	87	81

**Table 2 diagnostics-15-02540-t002:** Reported case of AML in pregnancy during the past years.

Author, Year	Study Type	Maternal Age at Diagnosis	GA at AML Diagnosis	Course of Pregnancy (TOP/WG at Delivery)	Chemotherapy Agents Used During Pregnancy	Fetal Outcome	Maternal Outcome
Yucebilgin 2004 [17]	Case report	24 years	29 WGs	Severe thrombocytopenia, acute fetal distress 33 GWs	IDA+ARA-C	Preterm delivery, Apgar score 2/6, favorable outcome	Alive, scheduled for remission–induction chemotherapy
Joseph 2006 [18]	Case report, 2 patients	No full text available	No full text available	No full text available	No full text available	No full text available	No full text available
Ticku 2013 [19]	Case report	22 years	26 WGs	Delivery 30 WGs, elective	HyperCVAD, postpartum several regimens changed	Premature delivery, ARDS	Alive, awaiting bone marrow transplant
Henig 2013 [20]	Systematic review, 120 AML, 53 APL	15–45 years	No distinct AML information	No distinct AML information	No distinct AML information	AGA weights, low incidence of malformations	Worse than non-pregnant individuals
Amiwero 2014 [21]	Case report, 1 AML	26 years	22 WGs	Delivery 30 WGs, spontaneous	NA	Preterm delivery	Alive
Boudry 2015 [22]	Case report, 1 AML	23 years	33 WGs	cesarean section, 33 WGs	NA	Preterm delivery	Death due to relapse 1 year later
Farhadfar 2016 [23]	Case series, 14 patients with AML	20–40 years	1st trimester—6	2 TOPs	1leukapheresis + busulfan 13—NA	Preterm delivery 1 SGA, ARDS 1 anemia Stillbirth 29 WGs	11 deaths (2 pregnant) 2 alive 1 status unknown
			2nd trimester—4	miscarriages 11 WGs, 17 WGs			
			3rd trimester—4	9 deliveries, 8 preterm stillbirths 29 WGs			

Frachiolla 2017 [1]	Case series, 5 AML	31–39 years	1st trimester—1	1 TOP	NA	3 preterm deliveries 1 stillbirth	5 CRs
			2nd trimester—1				
			3rd trimester—3	4 preterm deliveries			
Seegars 2017 [24]	Case report, 1 AML	26 years	17 WGs	35 WGs, induced delivery	DNR + ARA-C	Term delivery	No additional information
Mabed 2018 [25]	Case series, 18 AML	18–40 years	No distinct AML data		No distinct AML data	No distinct AML data	No distinct AML data
Horowitz 2018 [26]	Systematic review, 138 AML	No distinct AML data	No distinct AML data		No distinct AML data	No distinct AML data	No distinct AML data
Pineda-Mateo 2021 [27]	Case report, 1 AML	33 years	12+5 WGs	TOP	NA	-	Alive, awaiting bone marrow transplant
Zhu 2021 [28]	Case series, 14 AML	19–41 years	1st trimester—4	1 TOP, 1 miscarriage, 2 dead	2 DNR+Ara-C 4 DNR+Ara-C 1 IDA+Ara-C, NA	1 preterm delivery IUGR Favorable outcome	1 CR, 3 deaths
			2nd trimester—6	4 TOPs, 1 maternal death, 1 delivery			
			3rd trimester—4	2 preterm/2 term deliveries			
Diaz 2022 [29]	Case report, 1 AML	31 years	27 WGs	Elective cesarean section, 30 WGs	None	IUGR, died of early neonatal sepsis	Poor response to chemotherapy
Kumari 2022 [30]	Case report	18 years	33 WGs	Induced preterm delivery	Not reported	Preterm delivery Favorable outcome	Complete remission Relapse at 10 months and exitus during COVID-19 outbreak
Kriouille 2022 [31]	Case report	35 years	32 WGs	Preterm cesarean section	NA	IUGR	Not reported
Ding 2024 [32]	Case series, 25 AML	18–41 years	1st trimester—4	4 TOPs	2 cases, ATRA+DNR 1 case ATRA 1 case ATRA+ATO	13 deliveries 9 preterm/4 term 8 NICU admissions	7 PPHs 1 DICs 1 prenatal infection
			2nd trimester—13	8 TOPs, 5 deliveries			
			3rd trimester—8	8 deliveries			
Imane 2024 [33]	Case series, 1 AML	25 years	35 WGs	cesarean section, 36 WGs	NA	Good outcome	Complete remission
Kawtari 2025 [34]	Case report	25 years	35 WGs	Preterm cesarean section	AML03	Preterm delivery	CR
Ma 2025 [35]	Case series, 7 AML	19–37 years	1st trimester—1	1 miscarriage	2 IA 2 NA 2 HAA, hydroxyurea	3 preterm/1 term delivery 1 stillbirth	2 losses of follow-up 2 alive 3 deaths, 1 intrapartum
			2nd trimester—2	2 TOPs			
			3rd trimester—4	3 deliveries, 1 maternal death			
Cassidy 2025 [10]	Systematic review, 5 AML	31-39 years	1 first trimester	TOP	DNR+Ara-C DNR+Ara-C, NA DNR+Ara-C, NA	- 1 preterm 1 preterm/1 stillbirth	4 CRs, 1 death
			2 second trimester	2 deliveries			
			2 third trimester	2 deliveries			

GA—gestational age, AML—acute myeloid leukemia, TOP—termination of pregnancy, WG—weeks of gestation, NA—not applicable, ATRA—all-trans retinoic acid, DNR—daunorubicin, ATO—arsenic trioxide, NICU—neonatal intensive care unit, PPH—postpartum hemorrhage, DIC—disseminated intravascular coagulation, HAA—homoharringtonine, daunorubicin, and cytarabine, IA—idarubicin and cytarabine, IUGR—intrauterine growth restriction, SGA—small for gestational age, ARDS—acute respiratory distress syndrome, Ara-C—cytarabine, IDA—idarubicin, HyperCVAD—cytoxan, vincristine, adriamycin, dexamethasone, APL—acute promyelocytic leukemia, AGA—according to gestational age.

**Table 3 diagnostics-15-02540-t003:** Reported case of APL in pregnancy during the past 10 years.

Author, Year	Study Type	Maternal Age at Diagnosis (Years)	GA at APL Diagnosis	Course of Pregnancy/ GA at Delivery	Treatment	Fetal Outcome	Maternal Outcome
Puttirangsan 2025 [54]	case report	33	30 WGs	vaginal delivery, 34 WGs	ATRA+IDA (D2,4)	no complications	CR
Ni 2023 [55]	case report	31	13 WGs	cesarean section, 34 WGs	ATRA+ATO	no complications	CR
Li 2018 [56]	case report	24	25 WGs	maternal death, 25 WGs	ATRA+hydroxyurea	death	death
		37	34 WGs	vaginal delivery, 35 WGs	ATRA	no complications	CR
Nellessen 2017 [57]	case report	41	24 WGs	vaginal delivery, 34 WGs	ATRA before delivery and ATRA+ATO afterwards	no complications	CR
Li 2020 [56]	case report	28	28	vaginal delivery, 39 WGs	ATRA+ATO switched to ATRA+dauno	no complications	CR
Droc 2023 [58]	case report	27	17	stillbirth, unknown	ATRA+IDA	stillbirth	CR
Cochet 2020 [59]	case report	22	14	vaginal delivery, 40 WGs	ATRA+ATO	no complications	CR
Song 2015 [60]	case report	23	26	stillbirth, 28 WGs	ATRA	stillbirth	CR
Naithani 2016 [61]	case report	30	34	cesarean section, 35 WGs	No treatment before delivery	no complications	death in CR
Agarwal 2015 [62]	case report	40	26	cesarean section, 30 WGs	ATRA+IDA during pregnancy, ATRA+ATO post delivery	no complications	CR
Sisnawati 2024 [63]	case report	35	38–39	cesarean section, 38–39 WGs	Supportive treatment	alive, 3200 g	death
Dang 2020 [64]	case report	22	20	cesarean section, 32 WGs	ATRA+IDA	no complications	CR, later relapse
		25	25	caesarean section, unknown	ATRA+IDA+ATO		
Maruyama 2017 [65]	case report	30	23	death, 25 WG	ATRA+IDA	death, bilateral intraventricular hemorrhage	death
Khosla 2020 [66]	case report	23	35	vaginal delivery, 38 WGs	ATRA+ATO	no complications	CR
Zhang 2019 [67]	case report	23	25	cesarean section and death, unknown	ATRA	death	death
Siwatch 2024 [68]	case report	+20	unknown	Termination of pregnancy, unknown	ATRA+ATO	death	CR

ATRA—all-trans retinoic acid; IDA—idarubicin; D—days; ATO—arsenic trioxide; dauno—daunorubicine; unk—unknown.

## Data Availability

No new data were created or analyzed in this study. Data sharing is not applicable to this article.

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
