# Peer review of "An Account of Acute Myeloid Leukemia Complicating Pregnancy and Literature Review"

_diagnostics, 2025, doi:10.3390/diagnostics15192540_

Round 1

Reviewer 1 Report

Comments and Suggestions for Authors

The case report "An account of Acute Myeloid Leukemia Complicating Pregnancy and Literature Review" authored by Nemeti G et al is one of the rare occurrence of pregnancy and AML. It is an interesting read and well presented with the literature review. The figure one could be improved with removing the parenthesis and adding arrows (of different colors) instead. The authors could describe why the patient was not considered for FLT3 inhibitors despite the mutation positive status. Also authors should describe whether any assessment was performed on the foetus during pregnancy or after the birth for the chemotherapy related effects?

Author Response

Dear Reviewer,

Thank you very much for your careful and thoughtful review of our manuscript. We highly value your feedback and appreciate the time and effort invested into analysing our work.

We appreciate your suggestion, and we have worked to append the article accordingly:

Reviewer: The figure one could be improved with removing the parenthesis and adding arrows (of different colours) instead.

Authors: Thank you for your suggestion. We added the coloured arrows.

Reviewer: The authors could describe why the patient was not considered for FLT3 inhibitors despite the mutation positive status.

Authors: Thank you very much for pointing out this matter. Unfortunately, FLT3 inhibitors are not recommended in pregnancy. But currently she is receiving a FLT3 inhibitor and we have included this information in the article.

Reviewer: Also, authors should describe whether any assessment was performed on the foetus during pregnancy or after the birth for the chemotherapy related effects?

Authors: Thank you for your suggestion, it is true, we did not provide enough details regarding the fetal status. The following text was added to the case presentation:

“Fetal surveillance was normal at all prenatal check-ups, including biometry measurements, morphology scanning, Doppler studies and biophysical profile score.”

Reviewer 2 Report

Comments and Suggestions for Authors

The authors address the occurrence of AML during the first trimester of pregnancy and the management of a patient who did not consent to termination. However, the manuscript lacks essential clinical details:

  1. Maternal outcome: Please provide updated information on the mother. The delivery and the start of first-line therapy were reported on July 15, 2025. Now, in September, it is important to clarify whether the patient is alive, whether HSCT was performed, and what the current disease status is. These data are particularly relevant, since the Discussion section widely addresses this issue.

  2. Neonatal outcome: Apart from Apgar score and birth weight, no follow-up data on the infant are provided. At the time of submission, the child is already 4–5 months old. Please include information on:

    • complete blood count and peripheral smear,

    • organ ultrasound findings,

    • echocardiography,

    • overall growth and developmental milestones.
      Considering that the mother presented with hyperleukocytosis, it is also important to state whether blasts were present in the newborn’s circulation.

  3. Discussion section:

    • The subsection “AML diagnosis consideration” is not justified, as the authors did not investigate nor have data regarding congenital disorders in this patient.

    • Similarly, the paragraph and table concerning the treatment of APL in pregnancy should be removed, since the presented case is not APL.

    • In general, the Discussion includes multiple issues that cannot be related to the described patient. It should be re-edited to remain focused on the case.

  4. Definition of AML: Please correct the statement: “AML is characterized by the clonal proliferation of ≥20% poorly differentiated white blood cells, called blasts.” It should clearly specify that the threshold of ≥20% blasts refers to bone marrow.

  5. References: It is strongly recommended to cite the following recent and authoritative publications:

    1. Mills G, Shand A, Kennedy D, Lowe S, Bilsland V, Cutts B, McBride B, Brown W, Bolisetty S, Wegner EA, Kidson-Gerber G; Society of Obstetric Medicine of Australia and New Zealand; Haematology Society of Australia and New Zealand. Position statement on the diagnosis and management of acute leukaemia and aggressive lymphomas in pregnancy. Lancet Haematol. 2025; (e-pub ahead of print). doi: 10.1016/S2352-3026(24)00309-0.

    2. Ali S, Jones GL, Culligan DJ, Marsden PJ, Russell N, Embleton ND, Craddock C, BCSH Task Force. Guidelines for the diagnosis and management of acute myeloid leukaemia in pregnancy. Br J Haematol. 2015;170(4):487–495.

    3. Mills G, et al. New guidance on blood cancer during pregnancy. haematology.org. 2025 Jan 13.

Author Response

Dear Reviewer,

We would like to express our sincere gratitude for your thorough review of our manuscript. Your insights have been extremely valuable, and we greatly appreciate the time and effort you devoted to providing such detailed and constructive feedback.

The issues you identified—some of which were regrettably overlooked despite our collective review—have now been carefully addressed. We hope that the revised version meets your expectations. Please do not hesitate to let us know if any additional steps are required to further improve the manuscript.

Reviewer: Maternal outcome: Please provide updated information on the mother. The delivery and the start of first-line therapy were reported on July 15, 2025. Now, in September, it is important to clarify whether the patient is alive, whether HSCT was performed, and what the current disease status is. These data are particularly relevant, since the Discussion section widely addresses this issue.

Authors: Thank you very much for your suggestions. We did an update regarding the patient’s status.

Reviewer: Neonatal outcome: Apart from Apgar score and birth weight, no follow-up data on the infant are provided. At the time of submission, the child is already 4–5 months old. Please include information on complete blood count and peripheral smear, organ ultrasound findings, echocardiography, overall growth and developmental milestones.
Considering that the mother presented with hyperleukocytosis, it is also important to state whether blasts were present in the newborn’s circulation.

Authors: This suggestion is quite invaluable, thank you, our postnatal follow-up description was scarce and inadequate. The following was added to the respective part of the case presentation: “The fetal postpartum outcome was favourable with only mild, transient neonatal respiratory distress syndrome, and adequate weight gain. Blood tests did not reveal any hematologic abnormalities, regarding cell count or appearance on the blood smear. Internal organ ultrasound was normal, including echocardiogram. The baby was discharged in the 7th day of life, and reassessed at the 3 months follow-up, when development and neurologic acquirements were according to expectations and there were no pathologic findings at check-up analysis and imaging scans.

Reviewer: The subsection “AML diagnosis consideration” is not justified, as the authors did not investigate nor have data regarding congenital disorders in this patient.

Authors: Thank you for bringing up this issue, perhaps we can offer some clarification in support of our concept. Our patient benefited from genetic diagnosis to establish the AML subtype, such as any patient with leukaemia should, and this is detailed in the case presentation. On the other hand, this is the discussions section, and we thought it mandatory to run through the aspects facing the medical team in the case of a pregnant patient with AML from her clinical presentation and challenges thereof, to diagnostic considerations, management options, pregnancy outcome, neonatal outcome, and maternal long-term outcome. We feel this subchapter is appropriate in the general flow of the discussions, considering this is not only a case report, but also a literature review.

Reviewer: In general, the Discussion includes multiple issues that cannot be related to the described patient. It should be re-edited to remain focused on the case.

Authors: As stated above, throughout the discussions section we aimed to touch the important landmarks arising from pick-up to postnatal follow-up in the case of pregnant patients presenting with AML for the first time during their gestation. In each subchapter we touch our-case related information but provide insight from guidelines, as well as other cases, since management of disease in pregnancy is, in the end, a case-by-case approach according to the type of disease, gestational age at diagnosis, couple consent regarding pregnancy management/therapy, individual outcome. At the same time, national regulations on diverse matters and local availability of pharmacotherapeutic agents and blood products set their limits upon case management. When facing such a case, having access to literature describing similar case experiences provides valuable back-up.

Reviewer: Similarly, the paragraph and table concerning the treatment of APL in pregnancy should be removed, since the presented case is not APL.

Authors: Thank you for your comment, it is true, we were also undecided whether to include this information or not in the manuscript. However, since APL is a rare subtype of AML and there is no standardized approach to handling these patients, we opted to include this paragraph/table with the intent to provide both exhaustive information and a comparison with AML-proper cases. If you see fit, we will remove them.

Reviewer: Definition of AML: Please correct the statement: “AML is characterized by the clonal proliferation of ≥20% poorly differentiated white blood cells, called blasts.” It should clearly specify that the threshold of ≥20% blasts refers to bone marrow.

Authors: Thank you very much for your suggestion. The definition of AML is ≥20% blasts in the bone marrow or peripheral blood. We have rewritten the definition to be clear and correct.

Reviewer: References: It is strongly recommended to cite the following recent and authoritative publications.

Authors: Thank you for your valuable suggestion. The second citation was already included at number 7 and we were most glad to run through the information and include the latest reference you pointed out (suggestions 1 and 3 point to the same manuscript, the first one).

Reviewer 3 Report

Comments and Suggestions for Authors

I was presented with a case report for review concerning the diagnosis of AML in pregnancy. Below I will present my comments on the manuscript:

  1. In my opinion, the article has high clinical value because it presents a difficult case of AML with FLT3-ITD mutation. It has educational and practical value. It also clearly presents the clinical perspective regarding balancing the risks and safety for both mother and fetus.
  2. The abstract, in my opinion, is too long; the most important elements should be highlighted, and it should be indicated what this case report adds to the existing literature.
  3. In the further part of the presentation, details such as the echocardiography description or OGTT results could be presented in a table, which would reduce the volume of the text.
  4. In my opinion, the paper should include microscopic images of blasts as well as a detailed karyotype/molecular report.
  5. The discussion should be more structured; it could be divided into subsections, for example: first AML in pregnancy – diagnostic challenges and biological differences, then the FLT3-ITD mutation (its implications), followed by a section on treatment possibilities in pregnancy.
  6. It is necessary to explain why this particular treatment was chosen, even though, as far as I know, it is not the first-line treatment in pregnancy.
  7. In the tables, some entries contain “no full text available” or N/A.
  8. There are also some language errors, such as “than” instead of “then.”
  9. Abbreviations in the tables should be standardized.
  10. In the conclusions, it would be worthwhile to add something about azacitidine.

I believe the presented case has strong educational value; however, it is somewhat chaotic and needs to be better structured.

Author Response

Esteemed Reviewer,

We highly appreciate the time and effort invested in reading and critically assessing our paper. We appreciate your suggestions aimed to improve our work, please find below a point-by-point answer to your valuable inputs. Hopefully we have managed to improve the manuscript, and you will find it amenable to publication in the revised form.

Reviewer: In my opinion, the article has high clinical value because it presents a difficult case of AML with FLT3-ITD mutation. It has educational and practical value. It also clearly presents the clinical perspective regarding balancing the risks and safety for both mother and fetus.

Authors: Thank you for such a praised description of the aim of our endeavour, it is very gratifying to see our work understood and appreciated.

Reviewer: The abstract, in my opinion, is too long; the most important elements should be highlighted, and it should be indicated what this case report adds to the existing literature.

Authors: Thank you, we have brushed and appended the abstract as suggested. Its current form is also displayed below:

“Background and clinical significance: The occurrence of acute myeloid leukemia (AML) in pregnancy represents a diagnostic and management challenge in the attempt to balance and achieve both maternal and fetal wellbeing. Pregnancy specific manifestations mimic the initial symptoms of leukemia and may lead to a delay in diagnosis, especially during the first trimester of pregnancy. Decision-making strategies involve patient and couple counselling in a multidisciplinary team of hematologists, obstetricians, neonatologist and psychologist. Maternal outcome depends on the disease subtype, progression and response to medication. Fetal outcome depends on the potential other pregnancy complications, possible teratogenicity and gestational age at delivery, sometimes iatrogenic prematurity.

Case presentation: We present the case of a 38-year-old multiparous patient with a late first trimester AML diagnosis presenting for hyperemesis gravidarum-like symptoms. Genetic testing revealed the presence of a Fms like tyrosine kinase 3 - internal tandem duplication mutation (FLT3-ITD). Following repeatedly refused termination of pregnancy and with rapid disease progression azacitidine therapy was initiated. Elective caesarean delivery was performed at 34 weeks of gestation due to progressive leucocytosis which persisted postpartum requiring the use of first, second, and eventually third line chemotherapy. Fetal outcome was favorable at 3 months postpartum.

Conclusions: Cases of AML in pregnancy require a tailored approach according to guidelines but also patient/couple preferences, while the choice of chemotherapy is limited in light of its potential teratogenic effects. This is a case with misleading first presentation and challenging therapeutic choice due to its genetic subtype and maternal treatment postponing.”

Reviewer: In the further part of the presentation, details such as the echocardiography description or OGTT results could be presented in a table, which would reduce the volume of the text.

Authors: Thank you for your valuable comment. Upon re-reading, the description of the cardiac assessment was too stuffy, therefore we shortened it to “Repeated cardiologic assessment revealed normal cardiac appearance and function.” Is this new light we refrained from crafting a new table with the OGTT result, it seemed cleaner this way.

Reviewer: In my opinion, the paper should include microscopic images of blasts as well as a detailed karyotype/molecular report.

Authors: Thank you very much for you suggestions. An image of the karyotype was added. As the patient has normal karyotype, 46XX, we do not have any other relevant information on the report. A microscopic image of the blasts at day 15, after FLAG-Ida is already added in the article.

Reviewer: The discussion should be more structured; it could be divided into subsections, for example: first AML in pregnancy – diagnostic challenges and biological differences, then the FLT3-ITD mutation (its implications), followed by a section on treatment possibilities in pregnancy.

Authors: Thank you for your helpful comments. We tried to restructure some parts of the article and explained, hopefully better, why we chose this treatment approach, and which are the available, recommended, treatments at this moment.

Reviewer: It is necessary to explain why this particular treatment was chosen, even though, as far as I know, it is not the first-line treatment in pregnancy.

Authors: Thank you very much for your suggestion. You are right. We tried to better explain why this approach was chosen. Mostly, because this is what the patient wanted.

Reviewer: In the tables, some entries contain “no full text available” or N/A.

Authors: Thank you for pointing this out. We inputted “no full text available” when there was no method of obtaining the full-text article and “NA” – not applicable, when there was no information regarding the respective case (as sometimes in case series), or a certain aspect – choice of therapy, delivery route, outcome.

Reviewer: There are also some language errors, such as “than” instead of “then.”

Authors: Thank you, indeed. We corrected these errors and found a few more spell-checking issues which somehow escaped our repeated verifications.

Reviewer: Abbreviations in the tables should be standardized.
Authors: Thank you for pointing out this aspect, we have verified to keep the same abbreviations.

Reviewer: In the conclusions, it would be worthwhile to add something about azacitidine.
Authors: Thank you very much for your comment. The article has been updated on this matter.

Reviewer 4 Report

Comments and Suggestions for Authors

The article is a case report of acute myeloid leukemia (AML) in pregnancy with a FLT3-ITD mutation, followed by a literature review. Here’s a breakdown of what is original, relevant, and the specific gap it addresses in the field. A 38 year old pregnant patient diagnosed with AML during the first trimester. Since this is a single case report with literature review, the methodology is descriptive rather than experimental. Still, there are specific improvements and additional controls the authors could consider:

  1. The report describes maternal and neonatal outcomes, but a predefined set of outcome measures (maternal remission status, relapse, neonatal Apgar, growth, congenital anomalies, long-term follow up if available) would make the methodology more robust.
  2. Fetal monitoring controls. Serial ultrasound and Doppler assessments to systematically track growth restriction or placental insufficiency.
  3. A plan for long-term neonatal follow-up (neurodevelopment, growth) would strengthen conclusions about safety. Maternal fertility preservation or endocrine assessment could also be a relevant control.

In summary, while the study is methodologically solid with its prospective, controlled, and randomized design, integrating these improvements would significantly enhance the internal and external validity, reduce bias, and provide stronger evidence for clinical recommendations.

Author Response

Esteemed Reviewer,

We are grateful to the reviewer for the insightful and constructive comments provided. The observations have helped us to refine our manuscript and strengthen the presentation of our findings. We have carefully considered each point raised and made the corresponding revisions to improve the clarity, accuracy, and overall quality of the work.

Below, we address each comment in detail, outlining the modifications incorporated into the revised manuscript.

Reviewer:       The report describes maternal and neonatal outcomes, but a predefined set of outcome measures (maternal remission status, relapse, neonatal Apgar, growth, congenital anomalies, long-term follow up if available) would make the methodology more robust.

Authors: Thank you very much for your comment, iti is true that a single case report will lack proper methodology and we have appended the presentation of outcome measures for a better description of our case scenario. The information already provided has been emphasized. We have enhanced the fetal status evaluation, both antepartum and postpartum. However, we have kept it short since there were no pathologic findings and other reviewers have implied that we should refrain from being too stuffy.

“Fetal surveillance was normal at all prenatal check-ups, including biometry measurements, morphology scanning, Doppler studies and biophysical profile score.

(…)

The fetal postpartum outcome was favorable with only mild, transient neonatal respiratory distress syndrome, and adequate weight gain. Blood tests did not reveal any hematologic abnormalities, regarding cell count or appearance on the blood smear. Internal organ ultrasound was normal, including echocardiogram. The baby was dis-charged in the 7th day of life, and reassessed at the 3 months follow-up, when development and neurologic acquirements were according to expectations and there were no pathologic findings at check-up analysis and imaging scans.”

Reviewer: Fetal monitoring controls. Serial ultrasound and Doppler assessments to systematically track growth restriction or placental insufficiency.

Authors: Thank you for pointing out this aspect. Fetal surveillance has been performed throughout pregnancy with all the above-mentioned details and the manuscript has been changed. Fortunately, there were no pathologic aspects, therefore the description was kept to a minimum.

Reviewer: A plan for long-term neonatal follow-up (neurodevelopment, growth) would strengthen conclusions about safety. Maternal fertility preservation or endocrine assessment could also be a relevant control.
Authors: We appreciate your comment. Regarding neonatal care, we have added to the postpartum care section of our case presentation as presented above.

Regarding the fertility preservation direction, we only touched fertility issues at the end of discussions as “Fertility is affected by chemotherapy because it induces apoptosis in primordial follicles, which serve as the reservoir for future fertility. However, the risk of infertility increases with age and dosage and fertility preservation options counselling should be available for all patients. Younger patients may tolerate larger cumulative drug doses of chemotherapy and still maintain menstruation after a variable period [72, 73].” Also due to the fact, also included now in the same section, that out patient was 38 years old and delivering her fourth child.

Reviewer 5 Report

Comments and Suggestions for Authors

The article could improve its methodological rigor by adding transparency in diagnostic standards and literature review methods, by providing a structured framework for outcome reporting, and by including systematic maternal/fetal monitoring controls that can be benchmarked against guidelines.

  1. While the authors detail blood counts, smear, immunophenotyping, and genetic tests, the methodology section could be strengthened by explicitly mapping these steps to international diagnostic standards (WHO 2022 or ELN 2022 criteria) .
  2. The decision to use azacitidine instead of the conventional “3+7” regimen is explained, but it would help to provide structured justification (risk-benefit table for mother vs. fetus, reference to existing protocols in pregnancy).
  3. The review of reported AML-in-pregnancy cases (Table 2) is valuable, but methodology on how the literature was selected is missing. That makes it hard to assess completeness.

In summary, while the study is methodologically solid with its prospective, controlled, and randomized design, integrating these improvements would significantly enhance.

Author Response

Dear Reviewer,

We would like to sincerely thank you for the thorough evaluation of our manuscript and for providing feedback aimed at improving our work. We greatly appreciate the time and effort invested in helping reshape our article and improving its quality.

We have carefully read and implemented the changes suggested and hereby provide a point-by-point response to each comment.

Reviewer: The article could improve its methodological rigor by adding transparency in diagnostic standards and literature review methods, by providing a structured framework for outcome reporting, and by including systematic maternal/fetal monitoring controls that can be benchmarked against guidelines.

While the authors detail blood counts, smear, immunophenotyping, and genetic tests, the methodology section could be strengthened by explicitly mapping these steps to international diagnostic standards (WHO 2022 or ELN 2022 criteria).

Authors: Thank you very much for your comment, although, I must be truthful to say I do not entirely understand your point.

Regarding the international diagnostic standards, the criteria required for patient diagnosis according to WHO and ENL are highlighted in the article.

At the same time, although we had wanted to, when gathering the literature reports, it is impossible to have a common ground since every authors presented data in their own fashion, sometimes offering more details regarding cases, sometimes less, and sometimes providing information from different angles. I think that hematologic disease, pregnancy and the combination itself make case evolution and therefore management very unpredictable and unique and it is almost impossible to have a structured framework for outcome reporting.

Reviewer: The decision to use azacitidine instead of the conventional “3+7” regimen is explained, but it would help to provide structured justification (risk-benefit table for mother vs. fetus, reference to existing protocols in pregnancy).

Authors: Thank you very much for your valuable comments. We tried to elaborate and to explain why we chose this treatment approach, mostly this was influenced by the patients’ decision. References to existing protocols and toxicities both regarding the mother and fetus were added.

Reviewer: The review of reported AML-in-pregnancy cases (Table 2) is valuable, but methodology on how the literature was selected is missing. That makes it hard to assess completeness.

Authors: Thank you very much for your comment. The articles selected span 20 years back and include all articles (case series/case reports) published on this theme.

A descriptive phrase was included in the article just before Table 2, but it said 10 years, since that was our initial target. However, after finding quite a number of articles dating back, we decided to include them all and enlarge the tiem-frame.

“The caseload of pregnancies complicated by AML during the past 20 years is encompassed in Table 2 since there have been no national or global datasets available.”

Round 2

Reviewer 2 Report

Comments and Suggestions for Authors

Thank you for the responses and the authors’ explanations, both in the letter and in the text of the paper. The manuscript is ready for acceptance.

Author Response

Esteemed Reviewer,

Thank you for your reply and for the help with reshaping our work. Your comments have allowed us to add value to our manuscript.